# Germline *NUP98* Variants in Two Siblings with a Rothmund–Thomson-Like Spectrum: Protein Functional Changes Predicted by Molecular Modeling

**DOI:** 10.3390/ijms24044028

**Published:** 2023-02-16

**Authors:** Elisa Adele Colombo, Michele Valiante, Matteo Uggeri, Alessandro Orro, Silvia Majore, Paola Grammatico, Davide Gentilini, Palma Finelli, Cristina Gervasini, Pasqualina D’Ursi, Lidia Larizza

**Affiliations:** 1Genetica Medica, Dipartimento di Scienze Della Salute, Università Degli Studi di Milano, 20142 Milano, Italy; 2Laboratory of Medical Genetics, Department of Experimental Medicine, Sapienza University, San Camillo-Forlanini Hospital, 00152 Roma, Italy; 3Department of Biomedical Sciences National Research Council, Institute for Biomedical Technologies, 20054 Segrate, Italy; 4Department of Pharmacy, Section of Medicinal Chemistry, School of Medical and Pharmaceutical Sciences, University of Genoa, 16132 Genoa, Italy; 5Bioinformatics and Statistical Genomics Unit, IRCCS Istituto Auxologico Italiano, Via Ariosto 13, 20145 Milan, Italy; 6Department of Brain and Behavioral Sciences, University of Pavia, 27100 Pavia, Italy; 7Experimental Research Laboratory of Medical Cytogenetics and Molecular Genetics, IRCCS Istituto Auxologico Italiano, Via Ariosto 13, 20145 Milan, Italy; 8Department of Medical Biotechnology and Translational Medicine, University of Milan, 20133 Milan, Italy

**Keywords:** Rothmund–Thomson spectrum, juvenile cataracts, endocrine perturbation, whole exome sequencing, *NUP98* variants, FG repeats, intrinsic disordered regions, protein molecular modeling

## Abstract

Two adult siblings born to first-cousin parents presented a clinical phenotype reminiscent of Rothmund–Thomson syndrome (RTS), implying fragile hair, absent eyelashes/eyebrows, bilateral cataracts, mottled pigmentation, dental decay, hypogonadism, and osteoporosis. As the clinical suspicion was not supported by the sequencing of *RECQL4*, the RTS2-causative gene, whole exome sequencing was applied and disclosed the homozygous variants c.83G>A (p.Gly28Asp) and c.2624A>C (p.Glu875Ala) in the nucleoporin 98 (*NUP98*) gene. Though both variants affect highly conserved amino acids, the c.83G>A looked more intriguing due to its higher pathogenicity score and location of the replaced amino acid between phenylalanine-glycine (FG) repeats within the first NUP98 intrinsically disordered region. Molecular modeling studies of the mutated NUP98 FG domain evidenced a dispersion of the intramolecular cohesion elements and a more elongated conformational state compared to the wild type. This different dynamic behavior may affect the NUP98 functions as the minor plasticity of the mutated FG domain undermines its role as a multi-docking station for RNA and proteins, and the impaired folding can lead to the weakening or the loss of specific interactions. The clinical overlap of *NUP98*-mutated and RTS2/RTS1 patients, accounted by converging dysregulated gene networks, supports this first-described constitutional *NUP98* disorder, expanding the well-known role of NUP98 in cancer.

## 1. Introduction

The nuclear pore complex (NPC) is a large regulatory channel embedded in the nuclear envelope of the eukaryotes, primarily acting as gatekeeper for the nucleus–cytoplasmic transport of macromolecules, but is also involved in chromatin organization and regulation of gene transcription [1]. This proteinaceous machine is made up of 34 different nucleoporins (NUPs), evolutionarily conserved proteins, which are each present in multiple copies organized in distinct modules or sub-complexes: the outer transmembrane NUPs, the intermediately positioned scaffold NUPs, and the inner NUPs containing phenylalanine-glycine (FG) repeats [2], which line the central channel of the NPC [1]. The FG repeats, typically spacing 20 amino acids, represent intrinsically disordered regions (IDRs), which lack any secondary or tertiary structure conferring these domains a conformational dynamism that ensures extraordinary flexibility in binding and potential involvement in various protein–protein interaction networks [3]. It has been noted that the IDRs enrichment during evolution favored nucleoporins to achieve the multitasking function (“moonlighting proteins”) [4].

Beyond the canonical function of nucleo-cytoplasmic transport, several FG-NUPs (such as 98, 50, and 153) shuttle off the NPC to the nucleoplasm, where they can interact with chromatin modifiers and actively transcribed genes, participate in epigenetic and transcription regulation [5,6,7], and are involved in mitotic progression and DNA repair [8]. NUP98, residing on both the cytoplasmic and nuclear side of NPC, is a noteworthy representative of the ambivalence of a peripheral NPC component that is able to come off the nuclear pore [1,3].

NUP98 (OMIM*601021) [9] is an extremely disordered protein containing ~80% IDRs: the longest one (1–181) at the N-terminal region [2] is characterized by multiple FG (phenylalanine-glycine) and GLFG (glycine-leucine-phenylalanine-glycine) sequences. The FG motifs allow binding to different nuclear transporter receptors that regulate the bidirectional trafficking of proteins and RNA between the cytoplasm and the nucleus [10]. Inside the FG region, a small α-helical linker with a GLEBS (Gle2-binding sequence) motif mediates the interaction with the Ribonucleic Acid Export (RAE1) factor, a complex capable of binding single-strand RNAs [11]. The GLFG are necessary for the interaction in the nucleoplasm with the histone modifiers CBP/p300 and histone deacetylases [12,13].

The alteration of Nup98 expression has been associated with the dysregulation of hundreds of genes, such as tissue-specific *Hox* genes, and genes for the differentiation and maintenance of cell identity both in Drosophila and in mammalian cells [5,6,14]. Different studies in Drosophila showed that Nup98 in embryonic cells has a role in chromatin-binding and mediating enhancer-promoter looping of ecdysone-inducible genes [7], while in aged enterocytes, Nup98 is a subunit of a non-stop identity complex, which contributes to specific transcriptional programs related to cell identity, which, when damaged, lead to premature aging [15]. Further studies on NUP98 in human cells showed that it functions as a co-factor regulating the activity of DExH/DH-box helicase DHX9, Werner syndrome helicase, and other proteins of this family, which modulates gene expression and metabolism [16]. 

NUP98 is well-known for its recurrent somatic alterations in hematological malignancies, most often following gene fusions containing *NUP98* N-ter [17]. Conversely, no biallelic germline alterations associated with mendelian disorder(s) have been identified [3].

In the present work, we report the first observation of biallelic *NUP98* germline variants in two siblings presenting sparse eyebrows/eyelashes, bilateral cataracts, hypogonadism, skeletal defects, premature aging signs, a picture reminiscent of both Rothmund–Thomson syndrome type 2 (RTS2, OMIM#268400) [9,18,19], and RTS type 1 (RTS1, OMIM#618625) [9,20]. Patients were tested and found negative for *RECQL4* mutations, thus whole exome sequencing (WES) was applied in order to identify the causative gene that is mandatory to perform an adequate genetic counselling in genetic diseases characterized by highly variable clinical spectrum, such as RTS. WES on the affected siblings, their parents, and a healthy brother disclosed in both patients two homozygous highly linked non-synonymous variants of the *NUP98* gene, which was present in the heterozygous state in the healthy parents. To substantiate the molecular diagnosis, studies of protein molecular modeling were conducted on the *NUP98* variant with the higher pathogenicity score, which leads to the replacement of the highly conserved glycine 28 within the FG domain. Molecular dynamics simulations of the FG domain of the wild type and the mutated protein showed in the latter a decrease in the intramolecular cohesion elements, suggesting that changes in the NUP98 interactivity network might underpin the *NUP98*-related disorder.

## 2. Results

### 2.1. Clinical Report

Patient 1

The propositus III-1 (Figure 1a–c) is a 60-year-old man originating from Central Italy who was evaluated in the outpatient clinic of the Genetic Unit of San Camillo Hospital in Rome, concerning a syndromic condition suggestive of premature aging.

The delivery occurred a term after an uneventful pregnancy. Soon after birth, the presence of bilateral congenital cataracts was diagnosed. From the second year of life, the child’s eyelashes and eyebrows progressively became sparse and, in a short time, they were completely lost, a condition termed “madarosis”. At 8 and 14 years of age, he underwent sequential bilateral cataract surgery. Since the age of 12, he developed dental decay and, in a few years, became almost edentulous. Psychomotor development proceeded physiologically. At puberty, small and firm testicles were noticed and a condition of azoospermia was successively recognized. Starting from adolescence he manifested severe allergic reactions, comprising episodes of anaphylaxis to numerous foods (shellfish, banana, pineapple, and snails). At the age of 22, following trivial trauma, he suffered a patellar fracture, and osteoporosis was instrumentally detected by a dual-energy X-ray absorptiometry. At the age of 41, he underwent surgery for cluster hemorrhoids, while at the age of 44 he experienced a retinal detachment, which was surgically corrected. At the same age, due to dyspeptic symptoms, he underwent an endoscopic evaluation and was found to have multiple esophageal varices. At the age of 55, since biochemical evidence of very high serum ferritin and transferrin saturation percentage values, he underwent hematological and genetic evaluation, receiving the diagnosis of type 3 hereditary hemochromatosis (HH) (molecularly characterized). The same disorder was not present in his sister and other relatives. At the age of 60, his weight was 77 kg (75th centile), his height was 164 cm (3rd centile), his BMI was 28.6 (95th centile), and his Occipital-Frontal Circumference (OFC) was 57 cm (25–50th centile). At physical examination, the man showed a high forehead, proptosis, thin and fragile hair, sparse eyelashes (implanted), the absence of eyebrows (Figure 1a,b), the absence of hair in the axillary (Figure 1c) and pubic region, marked gynecomastia (reported to be present since adolescence), and hypoplastic testicles. The skin was dry (he reported scarce sweating). Several small hyperchromic patches were spread all over the body (Figure 1b,c), mainly on the back, face, and lower limbs. Some, described to have first appeared a few decades before, were keratotic. Adipose tissue showed a centripetal distribution and was poorly represented at the level of the lower limbs.

Hematochemical analyses showed persistently low testosterone levels (between 0.2 to 0.5 ng/mL) and elevated FSH (45.8 mIU/mL) and SHBG (81.8 nmol/L) values. LH was constantly normal. Mild transient anemia was also noted (Hb: 12.7; Ht: 39.9). Glucose, cholesterol, and triglycerides were within normal limits as well as renal function tests. Although the patient was affected by a quite severe form of HH and was treated with periodic phlebotomies, his hepatic enzymes and thyroid hormones resulted to be not altered. A bone mineral density test performed at the age of 53 reported a condition of severe osteoporosis (cervical spine T-score: −4.5, femoral T-score: −2.7) that worsened over the following years, despite vitamin D therapy. An MRI for the study of the pineal gland gave negative results.

Patient 2

The 57 years old proband’s sister, III-2 (Figure 1d–f) shares with his brother many clinical features with the exception of HH. She was found to have bilateral congenital cataracts that were surgically treated before adolescence. Similarly to her brother, the woman showed hair fragility since infancy and definitively lost eyelashes and eyebrows within the first years of life. Dental caries, progressively leading to the loss of numerous teeth, appeared before adolescence (she needed dental implants treatment). Menarche occurred at 13 years and the return cycles were scarce, tending to oligomenorrhea; she experienced natural menopause at 53 years. Since adolescence, the patient reported suffering from recurrent headache episodes and polyarticular joint pains worsening over the years. Since the age of about 30 years, she manifested depression and anxiety disorder symptoms controlled by specific medical therapy. From the third decade of life, she underwent several bone mineral density tests receiving the diagnosis of osteoporosis.

At first examination, the patient’s weight was 90 kg (>97th centile), her height was 163 cm (50th centile), her BMI was 33.8 (>97th centile), and her OFC was 54 cm (25–50th centile). A particularly high hairline was noticed (Figure 1e). Her scalp hair was thin and fragile, while her eyebrows were completely absent (Figure 1d). Her body hair appeared to be undetectable (Figure 1f). Likewise to her brother, she presented numerous aging patches, especially on the face, the upper part of the trunk, and the limbs. Keratotic lesions were also present. Her nails were normal. She presented bilateral eyelid ptosis.

At 45 years, gonadotropin dosage revealed low levels of FSH (1.9 mUl/mL) and normal LH (7.3 mUl/mL), while Estradiol-17β was at a high limit (296 pg/mL). Glucose, cholesterol hepatic enzyme, and triglycerides levels were within the physiological range. The renal function tests results were normal too. Cervical and lumbosacral MRI revealed multiple disc protrusions and herniation with endplate degenerative changes.

The clinical features of the affected siblings are summarized in Table 1.

Their consanguineous parents (first-degree cousins) and their younger brother are reported to be healthy.

The presence of thin and sparse hair, a lack of eyebrows/eyelashes, an early loss of teeth, bilateral congenital cataracts, endocrine problems, skeletal defects, and premature aging signs, including reduced bone density, raised the suspicion of Rothmund–Thomson syndrome type 2. RTS2 is an autosomal recessive disorder caused by *RECQL4* gene (OMIM*603780) mutations, which account for 60% of RTS cases [18,19]. *RECQL4* encodes the member Q4 of the RecQ family of ATP-dependent DNA helicases, which play a key role in genome maintenance and stability [21]. To perform genetic analyses, informed consent was obtained from all the participants who were enrolled in the study. As Sanger sequencing of *RECQL4* did not evidence any significant sequence alteration, genomic DNAs from the affected sibs (III-1 and III-2), their healthy brother (III-3), and parents (II-1, II-2) were processed by WES to disclose the molecular alteration underlying the syndromic condition.

### 2.2. Whole Exome Sequencing Discloses Single Nucleotide Variants of NUP98 Segregating from Parents to the Affected Siblings 

Filtering about 38.800 variants in each of the five loaded genomic DNA samples and prioritizing (under the hypothesis of a rare autosomal recessive condition) homozygous “identical by descent” variants transmitted by consanguineous parents to the affected siblings, allowed us to rule out the pathogenic alterations of *ANAPC1* gene (OMIM*608473), which were identified as causative of RTS1 [20]. *ANAPC1* encodes a component of the anaphase-promoting complex/cyclosome (APC/C) and its mutations account for 10% of clinically suspected RTS cases [20,22]. No alterations were disclosed in other candidate genes such as *WRN* and *BLM*: these genes encode RECQL3 and RECQL2 helicases, and their alterations are responsible for Werner (OMIM#277700) and Bloom (OMIM#210900) syndromes, which display clinical overlap with RTS [18,19]. Interrogation of the family reads evidenced in the affected siblings two strictly linked homozygous non-synonymous *NUP98* alterations, c.83G>A (p.Gly28Asp or p.G28D) in exon 3 and c.2624A>C (p.Glu875Ala or p.E875A) in exon 20 (Figure 2a–c). The exon 3 variant leads to the replacement of the non-polar glycine (G) by the polar charged aspartic (D) residue (Figure 2d) that falls within the de-structured region rich of FG repeats of NUP98 N-terminus. The exon 20 c.2624A>C variant causes the replacement of the glutamic acid (E) 875 by the apolar alanine (A), which is located twelve residues downstream the auto-proteolytic site of the pre-protein generating NUP98 (upstream) and NUP96 nucleoporins (Figure 2d) [23]. According to bioinformatics predictions, the post-translational cleavage should not be perturbed by this amino acid change. 

Both variants involve evolutionarily conserved amino acids (Figure 2d) and are predicted to be pathogenic or probably damaging by most consulted bioinformatics tools, with an overall score higher for exon 3 than exon 20 SNV (Single Nucleotide Variant) (Appendix A). The exon 3 SNV has not been described to date in relevant databases (Ensembl [24], 1000 genomes [25], GnomAD [26], Exome Variant Server [27]), being thus categorized as PM2 (pathogenic moderate) [28]. The exon 20 variant (rs760581747) has been detected in the heterozygous state at a very low frequency (MAF < 0.01) [24,26]. According to pPrint [29] and disoRDPbind [30] predictors, both the amino acids 29 and 876, following the residues replaced by exons 3 and 20 missense variants, are RNA-binding regions [2].

Sanger sequencing confirmed both missense variants in the two affected siblings, the carrier status of their parents (Figure 2c), and the homozygous condition for the wild type allele in the healthy brother.

The other biallelic homozygous variants highlighted by WES in other genes were all synonymous.

### 2.3. Mutated NUP98 Transcripts Escape mRNA Decay

Direct sequencing of the amplified cDNA fragments encompassing exons 3 and 20 SNVs on lymphoblastoid cell lines (LCLs) from the affected siblings revealed only transcripts carrying the sequence changes, demonstrating that the mutated transcripts do not undergo nonsense-mediated decay. Indeed, as attested by qPCR/real-time experiments, the expression levels of *NUP98* and *NUP96* transcripts in the patients’ lines were similar to those of healthy controls (data not shown). The parents’ samples were unavailable for this analysis. 

A transcript analysis performed on cDNAs from a panel of human tissues and cell lines confirmed that the *NUP98* gene is ubiquitously expressed (Appendix A).

### 2.4. NUP98 Variants Do Not Impair Histone H3K27 Acetylation

It has been shown that NUP98 interacts through its unique GLFG repeats with the CREB-binding protein CBP, a histone and non-histone acetyltransferase (KAT) [12]. To check the acetylation status of H3K27, a target of CBP, the AlphaLISA^®^ cellular assay, was performed on the protein samples obtained from LCLs of *NUP98*-affected sibs and from healthy controls, but no significant difference was observed.

### 2.5. NUP98-Mutated Cells Do Not Show Spontaneous Chromosomal Instability

Despite WES analysis excluded alterations in *RECQL4*, the RTS2-causative gene, and in genes associated with other RECQ helicase chromosomal instability syndromes, a preliminary analysis of chromosomal instability was performed on LCLs from both sibs and from control individuals. No signs of spontaneous chromosomal instability were highlighted by Giemsa staining, though the investigation needs to be repeated, possibly on fresh blood samples at the patients follow-up.

### 2.6. Molecular Modeling Studies of the NUP98 Variant

Though both missense variants involve highly conserved amino acids and are categorized as rare according to their <1% MAF, the p.G28D variant has been selected for studies of protein dynamics simulations for its higher predicted pathogenic score, with respect to the p.E875A variant (Appendix A), and its location in a linker segment between FG repeats within the first and longest (1–181 residues) NUP98 IDR. 

The FG domain at the N-ter of the NUP98 protein consists of 156 amino acids (aa). This domain is structurally disordered, and it is characterized by several FG sequence motifs.

The dynamic structure of the FG domain excludes it from classical structural biology analyses such as X-ray crystallography; thus, bioinformatics approaches could be used to elucidate the ensemble structures adopted by the wild type and variant proteins. Here, we used molecular dynamic (MD) simulations to characterize the dynamic ensemble of structures of the FG domain of wild type and mutated NUP98 proteins. Starting from a full-extended conformation of the FG domain (spanning residues 1–156), 20 replicas of MD simulations were performed for the wild type (WT) and the G28D variant, respectively. The structural diversity of each ensemble of biomolecular structures obtained by molecular dynamics simulations was analyzed by the Root-Mean Square Deviation (RMSD) values. The RMSD is a similarity measure that is widely used in the analysis of macromolecular structures and dynamics. It calculates the spatial difference between two conformational structures, which defines their conformational diversity. During the simulation, the RMSD values are calculated between the initial conformation of NUP98 and any of all the conformations of the succeeding frames. In a second time, the RMSD values are depicted as a line-style plot to find the convergence and stability of the simulation. In the present study, starting from a fully extended conformation model, the system passed over to a new spatial arrangement, which was stably maintained for 7 replicas in the WT (replicas 2, 4, 7, 11, 16, 17, and 18) and 15 replicas in the G28D variant (replicas 1, 2, 3, 5, 7, 8, 9, 10, 11, 13, 14, 15,16, 17, and 18) (Figure 3).

Based on these first results, it was possible to hypothesize that the WT and the G28D could explore different conformational spaces during their simulations. In particular, the mutated FG domain arrived at a minimum energy conformation more easily than the wild type, indicating that the G28D substitution could facilitate this process. To further pinpoint differences between the WT and the pathogenic NUP98 variant, an evaluation of the representative conformations of WT and G28D, obtained from a cluster analysis of the stable replicas, was carried out. 

#### 2.6.1. Compactness Differences Evaluation between the WT and the G28D

To verify the differences between the WT and the G28D FG domains, the compactness of the representative structures of the stable replicas was evaluated. It is known that, depending on the overall composition of the amino acids of the Intrinsically Disordered Protein (IDP), their compaction may differ, influencing the functional and bio-physical properties of the protein [31]. The compaction level and the shape of an IDP are assessed by the hydrodynamic radius, which is calculated by several biophysical experiments, such as dynamic light-scattering experiments, or they can be predicted from a computationally generated conformational ensemble of IDP, as performed in the present study.

The ensemble average of the hydrodynamic radius for the WT and G28D proteins was calculated, finding a hydrodynamics radius of 20.12 Å for the WT and of 20.69 Å for the mutated FG domain (Figure 4). These results suggested a different compaction for the two proteins. In particular, during the simulations, the initially fully extended conformations were shortened to globular conformations in the case of the WT, and they were more elongated in the case of the G28D.

#### 2.6.2. G28D Is Less Intramolecularly Cohesive Than WT NUP98

Since the G28D was found to be more expanse than the wild type, this provided a first indication that the variant is not as intramolecularly cohesive as the WT. It has been shown that phenylalanines in the FG motifs function as intramolecular cohesion elements imparting order to the FG domain and compacting its ensemble of structures in globular configurations [32]. To evaluate the intramolecular cohesion of the WT and mutated forms, we calculated the distances between the sites corresponding to the phenylalanine of the FG motifs (F-F pairs). The ensemble of structures of each stable replica of the wild type and mutated FG were sampled at intervals of 0.01 ns during the final 25 ns of each simulation and were used to calculate the distances between the F-F pairs. In the stable replicas, the average distance between the centers of mass of the phenylalanine residues belonging to the FG motifs is around 19 Å for the WT and 20.5 Å for the G28D (Appendix A). This observation highlights that the F-F pairs of the G28D are less close in comparison to the F-F pairs of the WT. Moreover, 23% of the G28D F-F pairs show a significative variation in the distance (>25%) in comparison to the WT (Figure 5).

To better describe the relationship between the FG motifs in the FG domain of the wild type and mutated NUP98, the distances of the F-F pairs in Figure 5 were used to generate a distance matrix, which was subsequently used to construct the network graphs. The network graphic shows the edges of the F-F pairs and the edge weights proportional to the distance value (Figure 6). 

These findings highlight that the intramolecular distances between phenylalanines of the FG motifs of the wild type and G28D differ quantitatively in the two proteins. In the wild type, phenylalanines are generally closer to each other, while in G28D, they are more scattered. This conclusion is consistent with the hypothesis that the FG motifs in the FG domain of NUP98 interact intramolecularly, in a manner similar to what was observed in other NUPs FG domains [33].

To assess the effect of D28 on intramolecular cohesions, the solvent accessibility of the replaced aspartic residue was calculated on the stable replicas of G28D by means of a Solvent-Accessible Surface Area (SASA). As shown in Appendix A, the aspartic can be exposed to the solvent or can be buried within the FG domain.

In the buried conformation, the D28 was found to make some common hydrogen bonds with specific residues of the protein: Gly30 residue in replicas 14 and 15, Thr31 in replicas 5, 14, and 15, Thr32 in replicas 7, 14, and 15, and Ser66 in replicas 2, 5, and 7. In these replicas, the D28 residue induced an arrangement around itself, driving some protein residues and FG repeats close to it. This arrangement prevented the appropriate interactions between the FG repeats, thus influencing the folding of the FG domain (Figure 7). Instead, when D28 was exposed, it did not allow a proper arrangement of the FG motifs around itself because of the chemical–physical properties of its side chain (steric hindrance and negative charge of the carboxyl group). In both conformations, D28 leads to a moving away of the intramolecular cohesion elements and a more elongated conformation compared to the wild type.

#### 2.6.3. RNA Interaction Impairment in the G28D Variant

As mentioned, the N-ter of NUP98 has the important functional role of multi-docking stations for RNA, proteins, and self-assembly [10]. In order to predict the possible differences in the interaction with the RNA between the WT and the G28D protein, the pPrint web server [29] was used to evaluate the residues involved in the RNA interaction. As shown in Appendix A, the results highlight that, in comparison to the WT, N26 and F37 of the G28D variant are predicted to lose their capability to interact with the RNA. It is worth noting that both N26 and F37 are close to the G28D mutation. Although this result suggests a minor ability of G28D to bind to the RNA, further analyses are needed.

Finally, it is possible to hypothesize that the G28D substitution may not allow the N-ter of NUP98 to be as dynamic as it should be, leading to protein misfunction, and possibly, impairing protein–protein and protein–RNA interactions.

## 3. Discussion

Mutations of *NUP98* have been so far investigated in cancer, mainly hematological malignancies, with the major focus on the expression of the *NUP98* chimeric allele. As regards germline variants, one case of a “heterozygous” mutation has been observed in a woman with a balanced translocation t (11;12), where a breakpoint in one chromosome 11 disrupts the *NUP98* gene [34] without causing any relevant constitutional phenotype as the carrier came to clinical observation due to bilateral renal angiomyolipoma.

We detected by WES the first germline *NUP98* “homozygous” likely pathogenic variants in two siblings with a clinical presentation that was reminiscent of Rothmund–Thomson syndrome. The siblings inherited from their heterozygous healthy parents the two linked biallelic SNVs in *NUP98* exons 3 (c.83G>A, p.G28D) and 20 (c.2624A>C, p.E875A), which affect evolutionarily conserved residues indicating functional relevance. Moreover, we attested that *NUP98*-mutated transcripts escape nonsense-mediated mRNA decay.

The higher predicted pathogenic score of exon 3 SNV (Appendix A) and the key position of the replaced amino acid between FG repeats within the first NUP98 IDR prompted us to characterize the FG domain of the G28D protein by molecular dynamics simulations to follow up its conformational motion over time, with respect to the WT FG domain. Our study evidenced a reduction in the G28D protein’s cohesiveness compared to the WT, which prevents the appropriate interaction among FG repeats. In addition, the D28 side chain can be exposed or buried to the solvent, thus influencing the folding of the FG domain and conferring a more elongated conformation in comparison to the wild type (Figure 7). We hypothesize that the different dynamic behavior of the NUP98-mutated FG domain could alter its interactions with the soluble nucleocytoplasmic transport machinery, RNA, proteins, and chromatin, causing a wide array of mutation-related epigenetic changes. Defects in the function of epigenetic regulators can drive changes in a myriad of genes with a pleiotropic phenotypic effect.

In keeping with the most reported NUP-caused human diseases [3], *NUP98* nucleoporopathy shows an autosomal recessive inheritance, which might suggest a loss-of-function mutation mechanism. However, *NUP98* belongs to the category of “goldilockers” genes, whose up or down misregulation disrupts normal function, accounting for the “paradoxical” result of cell-cycle exit with defects in protein synthesis that promote hallmarks of tumorigenesis observed in Drosophila by both Nup98 reduction and overexpression [35]. 

Till other *NUP98* variants inducing disease-relevant phenotypes are identified and further patients are characterized, we can only trace the gene networks candidate to be dysregulated based on the phenotypic signs shared by our *NUP98*-mutated siblings and individuals with the well clinically and molecularly characterized autosomal recessive RTS2 [18] and RTS1 [20]. The Venn diagram in Appendix A outlines the shared and unique features of the three clinical entities. Sparse hair, madarosis, skeletal defects, and premature development of aging are common to all three syndromes, with a major overlap of *NUP98*-related disease to RTS1 due to the shared bilateral juvenile cataracts and endocrine perturbation.

The RTS1 gene *ANAPC1* encodes the APC1 protein, the largest subunit of the anaphase-promoting complex (APC/C or cyclosome), which regulates the cell cycle, associating with either Cdc20 or CDH1 coactivators, and controls senescence, DNA replication, and DNA repair [36]. The APC has been shown to be essential for cell proliferation and differentiation in the lens in vitro [37], and mice heterozygous for *Anapc1* mutation have shown increased incidence of lens opacity [20]. RTS1 individuals who present the distinctive sign of bilateral cataracts have decreased APC levels and their synchronized fibroblasts in vitro undergo a defective cell cycle with a longer interphase [20]. Interestingly, it has been demonstrated that wild type NUP98 modulates the function of the APC/C to control mitotic entry, whereas it does not interact during mitosis [38] when the spindle assembly checkpoint safeguards proper chromatid separation [39]. NUP98 fusion oncoproteins whose expression induces mitotic spindle defects and chromosome mis-segregation, unlike the wild type NUP98, have been shown to physically interact with the Cdc20 APC/C regulator interfering with the APC/C function [38]. In addition, the nucleoporin NUP88 (OMIM*602552), residing on both the cytoplasmic and nuclear side of the NPC, associates with the NUP98-RAE1 sub-complex at the nuclear side and the NUP88- NUP98-RAE1 axis inhibits pre-mitotic activity of the anaphase-promoting complex/cyclosome [40]. Similar to NUP98, NUP88 is overexpressed in a variety of human cancers, and, in this state, it sequesters the NUP98-RAE1 complex in the cytoplasm away from APC/CDH1, disturbing mitotic checkpoint control and promoting aneuploidy [40]. Regulation of the cell cycle thus links ANAPC1 to NUP98.

The *NUP98* gene is also involved in multiple pathways of aging initiation: in Arabidopsis thaliana, *Nup98* mutation results in early senescence [41]. The *NUP98*-mutated siblings share bilateral cataracts and early onset of aging-associated diseases with the Werner prototypic aging syndrome (WS) [42]. In keeping with this connection, the evidence that NUP98 controls the localization and activity of the RNA helicase DHX9, which is regulated with opposite effect by the WS DNA helicase, might be meaningful [16]. 

WS on the other hand shows a marked overlap to RTS2, as the defects of both RECQL3 and RECQL4 DNA helicases, the caretakers of the genome, impair the essential pathways of DNA replication, recombination, repair, telomere maintenance, and chromosome segregation. Both syndromes are characterized by chromosomal instability leading to cancer [21]. 

RecQ helicases and nucleoporins have been defined as the guardians of the genome [43,44]. Emerging evidence points to the interactions between components of the nuclear pore complexes and proteins responsible for the maintenance of genome stability, as recently shown for FAM111A/B proteases and FG-NUPs [45]. It is worth noting that *FAM111B* (OMIM*615584) mutations cause POIKTMP (OMIM#615704), a syndrome entering in differential diagnosis with RTS2 and characterized by short telomeres-driven chromosomal instability [45,46]. Interconnected dysregulated gene networks are at the roots of the phenotypic similarities between NUP98 nucleoporopathy, RTS1, RTS2, RECQ helicase syndromes, and syndromes showing hallmarks of genomic instability. The phenotypic cross-syndromes comparison, with the limit of the numerical distortion of the clinical reports of >300 RTS2, 11 RTS1, and 2 here reported *NUP98*-affected individuals, is a powerful starting point as it may unveil the underlying converging pathways. Though all the above mentioned are ultra-rare diseases, the increasing use of omics approaches and the re-evaluation of unsolved patients with suggestive phenotypes may accelerate the pace of discovering the mechanistic basis of the NUP98 disorder and improve knowledge about NUPs off-pore functions on gene expression regulation. This knowledge will ultimately help to define the sequence of events leading to dysregulated developmental trajectories for the design of interventions and therapies improving the clinical outcome of the patients. 

The *NUP98*-related disorder adds to the list of rapidly expanding nucleoporopathies, confirming NUP98 as an integral player in mitotic events and gene expression regulation, and its alterations are responsible for constitutional diseases besides cancer. 

## 4. Materials and Methods

Clinical evaluation of two siblings raised RTS suspicion, but Sanger sequencing of *RECQL4*-causative gene did not disclose any alteration. WES processing of the index cases and family members evidenced in the siblings homozygous *NUP98* variants inherited by the healthy carrier parents. Pathogenicity of the missense variant affecting NUP98 FG domain was assessed by in silico DNA tools and protein molecular modeling. 

### 4.1. Patients

Two adult siblings, with a presumptive clinical diagnosis of Rothmund–Thomson syndrome, were referred to our laboratory by clinical geneticists (San Camillo Forlanini Hospital, University La Sapienza, Roma). Patients and family members were enrolled in the study after providing appropriate informed consent to genetic analysis and authorization to photos collection. The study protocol was approved by the Research Ethics Board of Istituto Auxologico Italiano, Milan, Italy, on 23/02/2016 (code 2016_02_23_17).

### 4.2. DNA Isolation and RECQL4 Analysis

Genomic DNA from peripheral blood was isolated with Wizard Genomic DNA Purification Kit (Promega) according to standard protocols.

The entire *RECQL4* gene (NG_016430.1) (21 exons and 20 introns with the exception of IVS12 minisatellite) was amplified and sequenced as previously described [47].

### 4.3. Whole Exome Sequencing and Data Analysis

WES was performed on 50 ng of genomic DNA from each member of the family (two affected siblings and their healthy parents and brother); the samples were prepared using the Illumina^®^ Nextera^®^ Rapid Capture Exome kit (Illumina, Hayward, CA, USA) and sequenced on Illumina HiSeq2500 sequencer (Illumina, Hayward, CA, USA). Burrows-Wheeler Alignment tool [48] was used to align the reads against the human reference genome (hg 19/GRCh37). SAM [49] and GATK [50] tools were employed to perform variant calling and annotation, respectively.

As the clinical phenotype of the two siblings and parents’ consanguinity pointed to an ultra-rare autosomal recessive disease, namely Rothmund–Thomson syndrome (RTS) or partially overlapping RECQ helicases syndromes, the variants list was filtered according to an allele frequency ≤0.1%, according to ExAC browser of Broad Institute [51] and 1000 Genomes [25] databases.

Beyond the autosomal recessive inheritance model, the type of variants (in order: nonsense, affecting the canonical splice-site regions, or non-synonymous) and the prediction of functional impact using Polyphen [52] and SIFT [53] tools were taken into account in the subsequent filtering steps.

### 4.4. Validation of NUP98 Variants

The candidate variants detected by WES within the *NUP98* gene were confirmed by Sanger sequencing of the tailored PCR amplicons. PCR encompassing both alterations was carried out with GoTaq DNA Polymerase (Promega, Milano, Italy) under standard conditions using the following primers: Fex3 5′-aatgccttttcatttggtcatctta-3′, Rex3 5′-ccagtgcttgtggaggtagc-3′, Fex20 5′-gagcaactagagcatacatcaa-3′, and Rex20 5′-tcaacttcggtatcacgga-3′. The amplicons were sequenced bidirectionally according to the manufacturer’s protocol using Big Dye Terminator v.3.1 Cycle Sequencing Kit (Applied Biosystems, Thermo Fisher Scientific, Waltham, MA, USA) on ABI PRISM 3130 Genetic Analyzer (Applied Biosystems). ChromasPro software 1.7.4 (Technelysium Pty Ltd. South Brisbane, Australia) was employed to analyze the electropherograms using the wild type sequence of the *NUP98* gene (NM_016320.5) as reference.

Description of sequence variants is according to HGVS recommendations [54]. The *NUP98* alteration was submitted to LOVD database [55].

### 4.5. Lymphoblastoid Cell Lines

EBV-transformed lymphoblastoid cell lines were established from peripheral blood lymphocytes of both affected sibs and from healthy controls.

LCLs were cultured in complete RPMI-1640 medium (EuroClone, Milano, Italy) supplemented with 10% fetal bovine serum (Lonza, Walkersville, MD, USA) and 1% penicillin, streptomycin, and ampicillin in a 37 °C humidified incubator with 5% CO_2_.

### 4.6. RNA Isolation, RT-PCR, and cDNA Analysis

Total RNA from LCLs was isolated using TRI Reagent (Sigma, St Louis, MO, USA) according to manufacturer’s protocols and then treated with DNase I (RNase-free, New England Bio-Labs, Inc., Ipswich, MA, USA). RNA concentration and purity/integrity were analyzed by NanoDrop spectrophotometer (Thermo Fisher Scientific, Waltham, MA, USA) and Agilent 2100 Bioanalyzer (Agilent Technologies, Palo Alto, CA, USA). High Capacity cDNA Reverse Transcription Kit (Applied Biosystems, Thermo Fisher Scientific, Waltham, MA, USA), and random hexamers were used to synthesize cDNA starting from 1000 ng of total RNA. Different fragments of *NUP98* transcript were amplified using GoTaq^®^ Flexi DNA polymerase (Promega, Milano, Italy) with the following primers F_ex2: 5′-atcatttggaacaccctttggg-3′, R_ex8: 5′-gttggccaaagagaccacct-3′, R_ex4: 5′-aaacccaaagccagtgcttg-3′, F_ex19: 5′-aataccggcctgaaactggt-3′, and R_ex21: 5′-actgtccagttccacaaccc-3′. Amplicons were run on 2% agarose gel and then sequenced as described above. 

Real-time PCR was performed in triplicates on QuantStudio Dx Real-Time PCR Instrument (Thermo Fisher, Waltham, MA, USA) with Fast SYBR^®^ Green Master Mix (Thermo Fisher, Waltham, MA, USA). The expression levels of *NUP98* transcripts were assessed using the previously described primers (F_ex2, R_ex4, F_ex19, and R_ex21), F_ex23 (5′-ggacgctcatttcgtgttgg-3′), and R_ex23 (5′-tgagctccagaggtgtctga-3′) encompassing the transcript region coding for NUP96 protein. 

Expression level of *GAPDH*, used as an internal control, was investigated using the following primers: F: 5′-acaacagcctcaagatcatcag-3′ and R: 5′- ggtccaccactgacacgttg-3′. The relative gene expression was determined using the ΔΔ Ct method [56].

### 4.7. Bioinformatic Prediction of Variants Pathogenicity

The bioinformatics tools Polyphen2 [52], SIFT [53], CADD [57], P-MUT [58], Panther [59], HOPE [60], Variant Effector Predictor [61], Mutation Assessor [62], Provean [63], and Mutpred2 [64] were applied to predict the effect of the identified variants at protein level.

Topfind [65], PeptideCutter-Expasy [66], and Procleave [67] were interrogated to uncover perturbations of the cleavage site.

### 4.8. Acetyl-Histone H3K27 Cellular Detection Assay

AlphaLISA^®^ cellular assay for detection of H3K27 acetyl marker was performed in accordance with the manufacturer’s protocols. Briefly, histones were extracted directly from the LCLs culture by adding lysis and extraction buffers. Next, a biotinylated anti-Histone H3 (C-terminus) antibody and AlphaLISA acceptor beads conjugated to an antibody specific to the marker that were incubated. Following 60 min at room temperature, streptavidin-coated donor beads were added and a final 30 min incubation was performed prior to plate reading. If acceptor and donor beads are neighboring, excitation of the donor beads with laser irradiation can be transmitted to acceptor beads triggering a chemiluminescent signal that is detectable with the EnSight instrument (PerkinElmer, Wolosem, MA, USA).

### 4.9. Chromosomal Instability Assay 

LCLs from both siblings and control adult individuals were analyzed for spontaneous chromosomal instability using Giemsa-stained chromosomal spreads. A total number of 30 metaphases per sample was scored for chromatid and chromosomal gaps and breaks.

### 4.10. Molecular Modeling Studies

The NUP98 wild type amino acid sequence spanning residues 1–156 was retrieved from UniProt [68] (Q9HDC8), and tleap software [69] was used to create a fully extended 3D structure of the wild type and G28D variation. Then, 20 replicas of 125 ns each of molecular dynamic simulation in gas phase were carried out for both the WT and the G28D.

The systems were minimized in 4 steps of minimization, for a total of 5000 steps of steepest descendent and 5000 steps of conjugate gradient with decreasing restraint, as follows:
A total of 1250 steps of steepest descendent, followed by 1250 steps of conjugate gradient with 5 Kcal/mol restraint on the whole system.A total of 1250 steps of steepest descendent, followed by 1250 steps of conjugate gradient with 3 Kcal/mol restraint on the whole system, except for the hydrogens.A total of 1250 steps of steepest descendent, followed by 1250 steps of conjugate gradient with 1 Kcal/mol restraint on the whole system, except for the hydrogens.A total of 1250 steps of steepest descendent, followed by 1250 steps of conjugate gradient without restraints.

Subsequently, starting from the minimized structure, 20 independent replicas with different seeds of 125 ns each were carried out on both systems, without any restraint. The ff14SB forcefield was used for the simulation, with a salt concentration of 0.15 mM. The MD trajectories analyses, RMSD, F-F pair distances, and SASA, were carried out using the Ambertools package. The network graph was created using the R-Studio [70].

The hydrodynamic radius was calculated using the HullRad webserver [71] and the residues interacting with the RNA have been predicted using Pprint webserver with defaults settings [29].

## 5. Conclusions

To date, the *NUP98* gene has been studied for its involvement in a spectrum of hematopoietic malignancies, mainly pediatric leukemias, where chromosomal translocations give origin to chimeric oncoproteins, presenting the intrinsically disordered N-terminal region of NUP98 fused with over 30 different partners.

We report the first germline homozygous *NUP98* alterations, c.83G>A (p.G28D) and c.2624A>C (p.E875A), identified by WES in two adult siblings who present multiple clinical signs recalling Rothmund–Thomson syndrome. Both missense variants affect highly conserved amino acids; however, the c.83G>A is here in the limelight due to its higher pathogenicity score and location of the replaced amino acid between FG repeats in the first NUP98 IDR. After confirming the presence of only mutated transcripts in the affected siblings, we performed molecular modeling studies to investigate whether and how the change in glycine 28 to aspartic might influence the structure and function of the NUP98 protein. The analysis revealed a reduction in the mutated protein of cohesiveness influencing its folding and the flexibility required for NUP98 “pore” and “off-pore” interactions and functions. 

The different dynamic behavior of mutated NUP98 may compromise its role as a multi-docking station for proteins and RNA, and as hub for the recruitment of chromatin and nuclear factors needed for the maintenance of genome stability.

The link between the *NUP98* alterations and the observed phenotype is supported by the data we obtained in silico and by the emerging evidence that connects *NUP98* to genes and pathways of syndromes with partial phenotypic overlap and hallmarks of genomic/chromosomal instability.

## Figures and Tables

**Figure 1 ijms-24-04028-f001:**
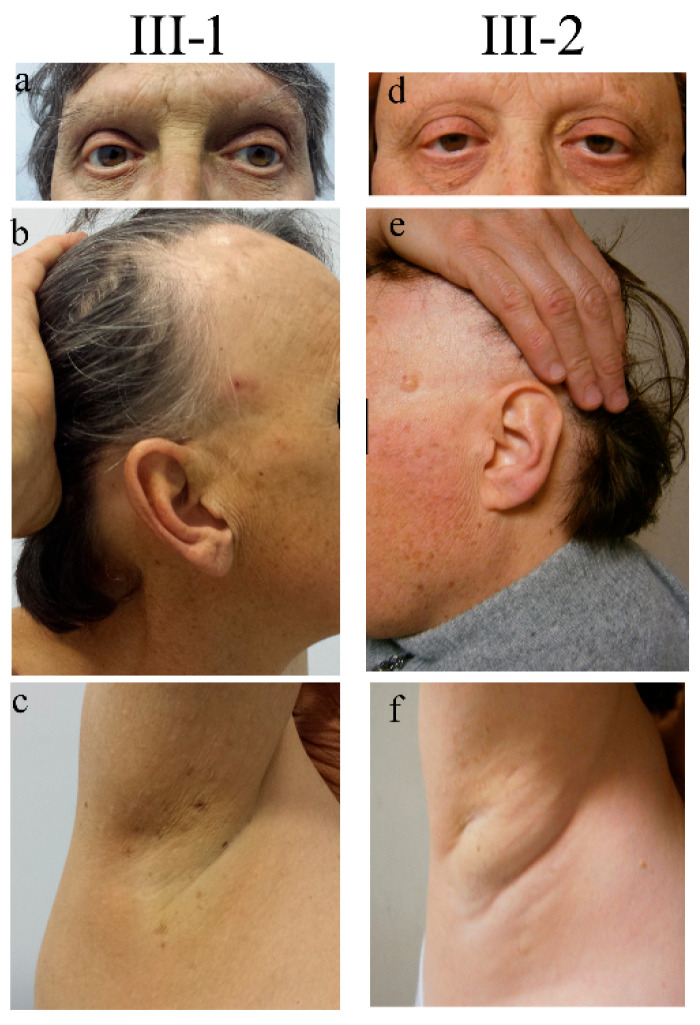
Clinical signs in two siblings carrying *NUP98* alterations. Representative pictures documenting the same signs both in the male (III-1) and the female (III-2) patients: ptosis and absence of eyelashes and eyebrows (**a**,**d**), sparse hair and mottled pigmentation on the face (**b**,**e**), and absence of ancillary hair (**c**,**f**). Pictures were taken when patients III-1 and III-2 were 55 and 52 years old, respectively.

**Figure 2 ijms-24-04028-f002:**
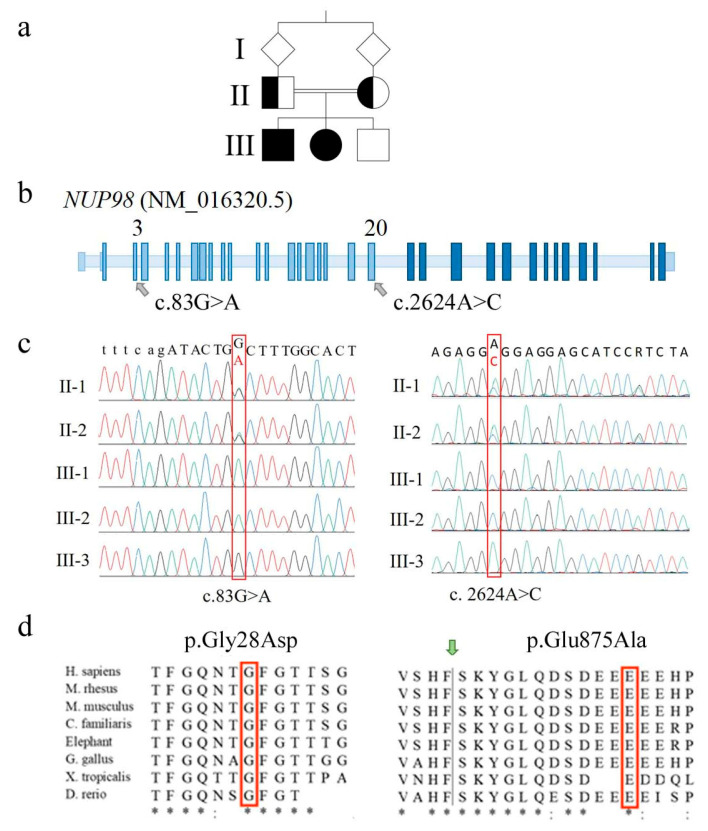
Molecular identification marks of *NUP98* alterations detected in the family. (**a**) Pedigree of the family with two affected siblings (III-1 and III-2, filled square and circle) whose parents are first-cousins. (**b**) Positioning of the two identified *NUP98* alterations onto the schematic genomic structure of the *NUP98* gene (not in scale). Light blue boxes indicate exons coding for NUP98 protein while dark blue ones indicate exons coding for NUP96 protein. (**c**) Electropherograms showing the two alterations (red frame) identified in homozygous state in the affected siblings and in heterozygous state in the parents. To note, both variants are not detected in the healthy brother III-3, suggesting they are highly linked on the same allele (cis). (**d**) Phylogenetic conservation of the two fragments with the amino acid residues affected by the identified sequence changes in the red frame. Identical amino acids are asterisked; highly similar residues are indicated by semicolons. The green arrow indicates the auto-proteolytic cleavage site of the precursor protein allowing the separation of NUP98 from NUP96.

**Figure 3 ijms-24-04028-f003:**
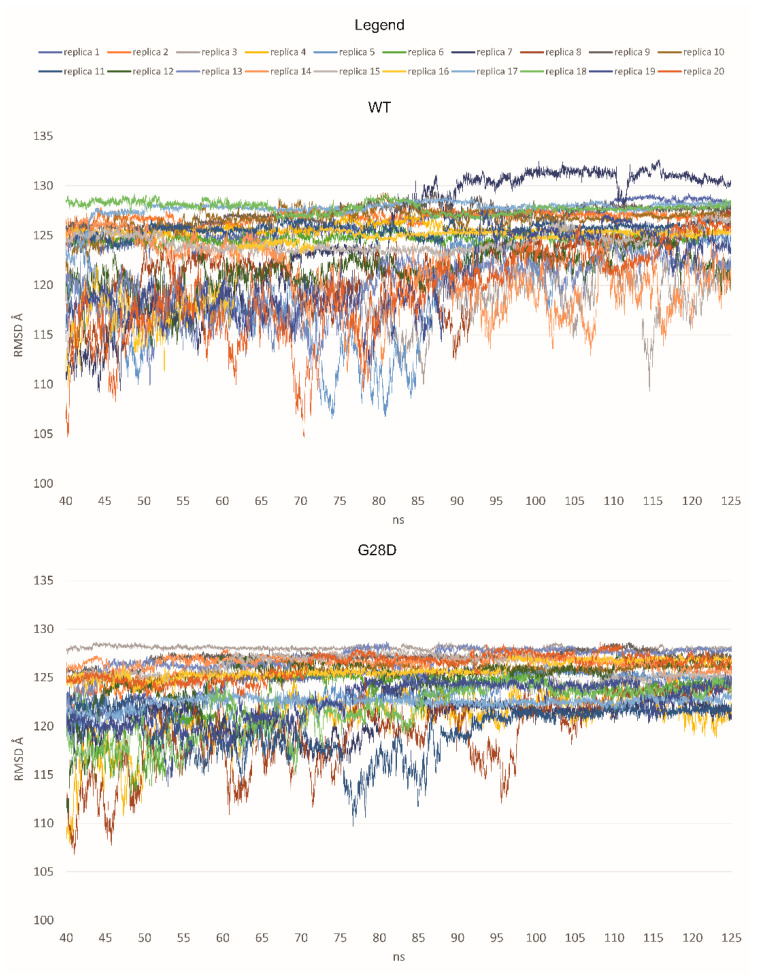
RMSD values for each replica of the WT and the G28D of NUP98 FG domain. In the plot, the RMSD values are shown from 40 nanoseconds (ns) of simulation to make the observation of the stable replicas easier.

**Figure 4 ijms-24-04028-f004:**
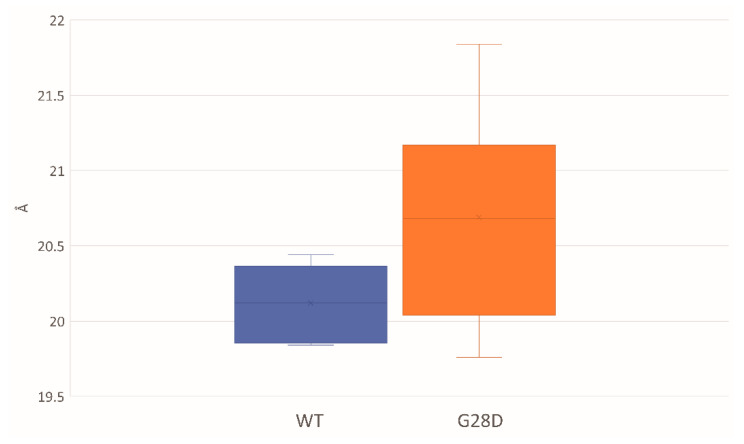
Hydrodynamic radiuses of the WT and G28D NUP98 FG domain from MD stable replicas. Box-plot of the average of the hydrodynamic radius that is expressed in units of Angstroms and is calculated from the representative structures of WT (blue) and NUP98-mutated (orange) replicas that have reached equilibrium.

**Figure 5 ijms-24-04028-f005:**
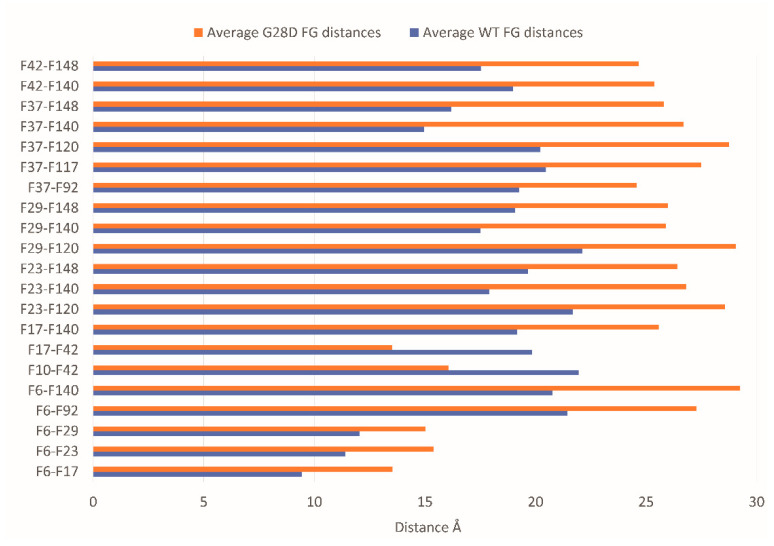
The distances of the G28D F-F pairs. Distance between the centers of mass of the F-F pairs belonging to FG motifs in WT (blue bars) and G28D (orange bars) NUP98 FG domain. In the plot are reported only the G28D F-F pairs with a variation in distance >25% compared to wild type NUP98.

**Figure 6 ijms-24-04028-f006:**
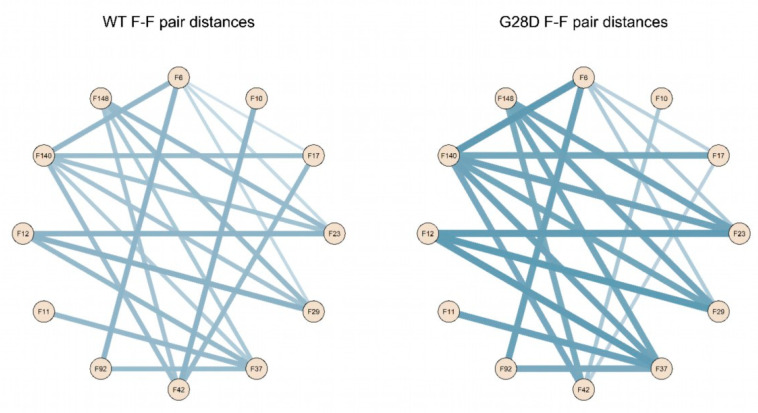
Schematic representation of F-F pair distances in the wild type and G28D. Intramolecular cohesions in the simulated FG domains are shown as a schematic representation of F-F distances in the wild type and G28D with lines whereby the thickness increases with the distance.

**Figure 7 ijms-24-04028-f007:**
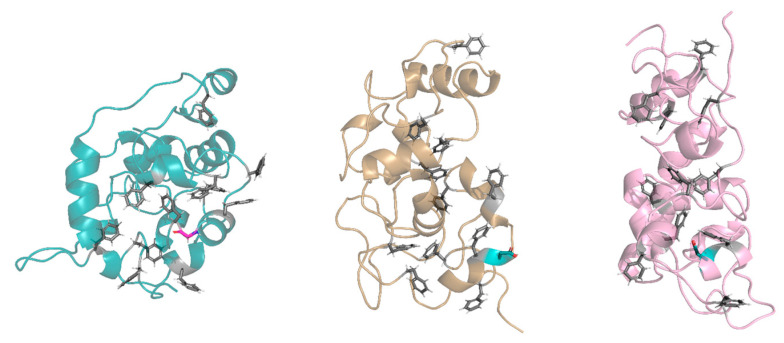
Exposed and buried conformations of G28D in comparison with wild type. On the left, a conformation of the wild type (azure ribbon), in the centers, a conformation of the G28D with the D28 in the exposed conformation (orange ribbon), and on the right, a conformation of the G28D with the D28 in the buried conformation (pink ribbon). Phenylalanines of the FG motifs are shown in grey sticks, while the G28 and the D28 are in purple and in cyan stick, respectively.

**Table 1 ijms-24-04028-t001:** Overall clinical phenotype of the affected siblings.

Tissue/Organ/ Apparatus	III-1	III-2
Skin Hair	Mottled pigmentation since adolescence Dry skin Thin and fragile hair Absent eyelashes since infancy Absent eyebrows since infancy	Mottled pigmentation since adolescence Fragile hair Absent eyelashes since infancy Absent eyebrows since infancy
Teeth	Dental decay since adolescence	Dental decay since adolescence
Eyes	Bilateral cataracts since infancy Retinal detachment in adulthood	Bilateral cataracts since infancy
Bones	Osteoporosis in third decade	Osteoporosis in third decade Polyarticular joint pains
Endocrine system	Hypogonadism Azoospermia Gynecomastia	Oligomenorrhea
Blood	Hereditary hemochromatosis type 3	-
Other	Short stature Allergies to several foods Multiple esophageal varices Constipation	Eyelid ptosis Recurrent headache Depression and anxiety disorder

## Data Availability

The *NUP98* variant has been submitted to the LOVD database.

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
