# Peer review of "Germline NUP98 Variants in Two Siblings with a Rothmund–Thomson-Like Spectrum: Protein Functional Changes Predicted by Molecular Modeling"

_ijms, 2023, doi:10.3390/ijms24044028_

Round 1
Reviewer 1 Report
Introduction:
1. Authors can describe the urgency of this study
2. Authors can explore biological plausibility
Methods:
1. Authors can describe design study
Referenences:
1. Authors can provide references not more than 10 years old
Author Response
We thank Referee 1 for his/her suggestions.
Point 1: Authors can describe the urgency of this study
Response 1: Thank you for raising this important point. Patients with a genetic disease should receive a molecular diagnosis for genetic counseling and appropriate managing of medical complications. Looking for the “culprit” gene by Whole Exome Sequencing we found in both affected siblings homozygous variants in Nup98, a gene known to be often rearranged in cancer, but not yet involved in constitutional disorder. For this reason, we have now underlined in the introduction: “Whole exome sequencing (WES) was applied in order to identify the causative gene that is mandatory to perform an adequate genetic counselling in genetic diseases characterized by highly variable clinical spectrum as RTS”.
To explore the significance of the homozygous variants detected by WES, we added: “To substantiate the molecular diagnosis, studies of protein molecular modelling were conducted on the NUP98 variant with the higher pathogenicity score”.
At the end of the Discussion, we foresee that “the increasing use of omics approaches and the re-evaluation of unsolved patients with suggestive phenotype may accelerate the pace of discovering the mechanistic basis of NUP98-disorder. This knowledge will ultimately help to define the sequence of events leading to dysregulated developmental trajectories for the design of interventions and therapies improving the clinical outcome of the affected patients”.
Point 2:1 Authors can explore biological plausibility
Response 2: We thank the reviewer for the insightful suggestion to explore the biological plausibility that homozygous variants in NUP98 might be responsible for the RTS-like phenotype of our index cases. We added to the arguments of the shared gene networks of NUP98, ANAPC1 and RECQL4 which account for the partial overlap of NUP98-disorder with RTS1 and RTS2 syndromes, the emerging evidence in the literature on the interactions between nuclear pore complexes and proteins responsible for the maintenance of genome stability. We mentioned in the Discussion the interaction of FAM111 proteases with FG nucleoporins pointing out that: i) FAM111B is encoded by a gene which mutation is responsible for POIKTMP, a syndrome entering in differential diagnosis with RTS2 and ii) FAM111B-mutated cells show hallmark of genomic instability.
Point 3: Authors can describe design study
Response 3: We thank the reviewer for proper indication and added as first paragraph of the Methods the study design which shortly delineates the sequence of the experimental steps which are thereafter detailed.
Point 4: Authors can provide references not more than 10 years old
Response 4: According to the reviewer indication to give priority to the recent literature, we have deleted several old references and have maintained just a few landmark references which provide the background to our study and the context for discussion.

Reviewer 2 Report
This is a very good written and informative manuscript.
1- I suggest you add that your article reviews the subject in the title.
2- Consider summarizing the introduction and discussion sections if possible.
3- The number of keywords seems too high.
Author Response
We thank the reviewer for appreciation and favorable comments on our manuscript.
Point 1: I suggest you add that your article reviews the subject in the title.
Response 1: We added a few-line in the Conclusion section where, as suggested by the reviewer, we underline that we have shown and discussed the subject anticipated by the title.
Point 2: Consider summarizing the introduction and discussion sections if possible.
Response 2: We admit that the introduction has been the most difficult part of the manuscript to write due to the exceedingly wide literature on nucleoporins and their multiple functions. However, we compacted it as much as possible. We also deleted/summarized some sentences in the Discussion.
Point 3: The number of keywords seems too high.
Response 3: We thank the reviewer for the correct observation and reduced the number of key words as recommended.
Reviewer 3 Report
Thank you for your trust and entrusting the role of a reviewer. Due to the constantly increasing number of chromosomal disorders, new disease entities appear that often have features of several syndromes. The increasing possibilities of genetic diagnostics create the conditions for diagnosing rare syndromes. Rothmund-Thompson syndrome is a developmental disorder characterized by premature aging. Changes occur in early childhood and progress. We cannot treat it, but we can bring this disease entity closer to practicing doctors. This allows for symptomatic and supportive treatment with a team of specialists. The paper suggests that in the case of close kinship of the parents, genetic tests would be recommended as to the possibility of chromosomal disorders in the offspring.
Author Response
We thank the reviewer for appreciating our work. We fully agree on the accelerated diagnosis of rare/ultrare diseases thanks to refinement and update of the diagnostic toolkit. Clinical and molecular data of our work are consistent with the reviewer’s opinion that often the new genetic entities have features of several known syndromes.
Reviewer 4 Report
Rothmund-Thomson syndrome (RTS) is an autosomal recessive genodermatosis characterized by a rash that progresses to poikiloderma; sparse hair, eyelashes, and/or eyebrows; small size; skeletal and dental abnormalities; juvenile cataracts; and an increased risk for cancer, especially osteosarcoma. RTS is a very rare disease and reliable data on its prevalence are not available, only a few hundred are recorded. This manuscript certainly adds value to the clinical awareness of RTS. The diagnosis of RTS can be difficult due to its highly variable clinical spectrum. Molecular diagnosis is the only available tool to a subgroup of RTS patients that can provide them with a basis for adequate genetic counseling. Patients should be managed by a multidisciplinary team and offered long-term follow-up.
Author Response
We thank Referee 4 for his/her appreciation and favourable comments on our work. It is perfectly true that RTS has a highly variable clinical spectrum and only molecular diagnosis permits subgrouping of patients and adequate counselling and management. Our study adds a small plug to this awareness.
Reviewer 5 Report
30th January 2023
The paper entitles as “Germline biallelic NUP98 variants in two siblings presenting a Rothmund-Thomson like spectrum: functional changes borne out by protein molecular modelling studies” by Elisa A. Colombo et al. They report a gene variant which behavior may affect the role of NUP98. Mutations of NUP98 have been so far investigated in cancer. Here they detected by WES the first germline NUP98 “homozygous” likely pathogenic variants in two siblings with a clinical presentation reminiscent of Rothmund Thomson syndrome. This syndrome is a rare, inherited disorder that affects many parts of the body, especially the skin, eyes, bones, hair, and teeth. Therefore, the connection is very clear due to people with Rothmund-Thomson syndrome have an increased risk of developing cancer, particularly a form of bone cancer called osteosarcoma.
The novelty of this paper is clear do to approximately 2/3 of individuals with RTS are found to have an abnormality (mutation) in the RECQL4 gene however Elisa A et all found that variants in NUP98 gen modifying the conformational folding can produce the disease.
Methodology is clear and very well described and I do not have further recommendations
I only have one minor recommendations
· Figures 3 and 5 are not eligible.
· The abstract is not very informative and hard to reed I recommend a profound revision
· I think the title can be improved.
· Figure 4 the labelling of the figure doesn’t correspond with the figure footnote.
Author Response
Point 1: Figures 3 and 5 are not eligible.
Response 1: We thank the reviewer for the observation and recommendation; we agree that Figures 3 and 5 are not legible. Thus, we remade the figures. In particular, Figure 5a has been inserted in the supporting information as S2, while Figure 5b has remained as Figure 5 in the main text (which has been accordingly modified).
Point 2: The abstract is not very informative and hard to read I recommend a profound revision
Response 2: We thank the reviewer for the recommendation and deeply revised the abstract which should be now more clear and informative.
Point 3: I think the title can be improved.
Response 3: We thank the reviewer for important advice and modified the title attempting to better convey the core message of our work. The new title is “Germline NUP98 variants in two siblings with a Rothmund-Thomson-like spectrum: protein functional changes predicted by molecular modelling”.
Point 4: Figure 4 the labelling of the figure doesn’t correspond with the figure footnote.
Response 4: We thank the reviewer for the observation: Figure 4 and footnote have been corrected.